# A Novel Method to Assess Motor Cortex Connectivity and Event Related Desynchronization Based on Mass Models

**DOI:** 10.3390/brainsci11111479

**Published:** 2021-11-08

**Authors:** Mauro Ursino, Giulia Ricci, Laura Astolfi, Floriana Pichiorri, Manuela Petti, Elisa Magosso

**Affiliations:** 1Department of Electrical, Electronic and Information Engineering Guglielmo Marconi, Campus of Cesena, University of Bologna, Via Dell’Università 50, 47521 Cesena, Italy; giulia.ricci29@unibo.it (G.R.); elisa.magosso@unibo.it (E.M.); 2Department of Computer, Control and Management Engineering, Sapienza University of Rome, Via Ariosto, 25, 00185 Roma, Italy; laura.astolfi@uniroma1.it (L.A.); manuela.petti@uniroma1.it (M.P.); 3Fondazione Santa Lucia, IRCCS Via Ardeatina 306/354, 00179 Roma, Italy; f.pichiorri@hsantalucia.it

**Keywords:** EEG, motor cortex after stroke, network model, model–based connectivity, non–linear coupling, excitatory/inhibitory synaptic connections, brain rhythms

## Abstract

Knowledge of motor cortex connectivity is of great value in cognitive neuroscience, in order to provide a better understanding of motor organization and its alterations in pathological conditions. Traditional methods provide connectivity estimations which may vary depending on the task. This work aims to propose a new method for motor connectivity assessment based on the hypothesis of a task-independent connectivity network, assuming nonlinear behavior. The model considers six cortical regions of interest (ROIs) involved in hand movement. The dynamics of each region is simulated using a neural mass model, which reproduces the oscillatory activity through the interaction among four neural populations. Parameters of the model have been assigned to simulate both power spectral densities and coherences of a patient with left-hemisphere stroke during resting condition, movement of the affected, and movement of the unaffected hand. The presented model can simulate the three conditions using a single set of connectivity parameters, assuming that only inputs to the ROIs change from one condition to the other. The proposed procedure represents an innovative method to assess a brain circuit, which does not rely on a task-dependent connectivity network and allows brain rhythms and desynchronization to be assessed on a quantitative basis.

## 1. Introduction

Any movement is the result of the interaction among several brain regions, which are mutually interconnected via excitatory and inhibitory links, and whose interplay governs motor preparation and execution. Understanding how these regions work together and establishing a reliable connectivity network is a matter of intense study in neuroscience today, to improve our comprehension of different aspects of motor behavior. Moreover, it is well known that this motor network is altered in pathological conditions, particularly after stroke, leading to abnormal interactions not only in the lesioned area but also in remote regions [1,2,3].

A wealth of studies in recent years attempted to quantify the motor network in both healthy subjects and patients, using model–based approaches: in these studies, connectivity is estimated starting from an explicit model of causal inference, usually expressed in terms of state space differential equations ([4,5,6,7,8]). Most previous works use a bilinear state space model (see [5] which incorporates an intrinsic (i.e., task–independent) connectivity matrix, a task–dependent connectivity matrix, explaining the changes in neuronal states during the respective task, and a matrix for the experimental inputs that drive regional activity).

Besides works that investigated motor network in health and disease using causal models, other inferred connectivity by using data-driven approaches, such as correlation, coherence, Granger causality, direct transfer function, partial directed coherence (PDC), mutual information or transfer entropy (TE) ([9,10,11,12]). In general, model-based and data-driven methods can be considered as complementary approaches due to their respective advantages and limitations, and the comparison between their results can provide a deeper understanding of brain functions.

A typical result of the studies mentioned above is that the estimated connectivity network changes as a function of the performed task. Generally, during unimanual movements, connectivity towards the contralateral primary motor cortex (M1) is increased, whereas connectivity towards the ipsilateral motor areas is reduced [5,13]. Performing hand movements at a higher frequency is associated with a linear increase in connectivity strength [7]. Differences are also evident by comparing motor execution vs. motor imagery for the same task: in these cases, modifications involve both some connectivity weights in the premotor cortices (PMCs) and supplementary motor areas (SMAs), and the inputs to these regions [4,14].

Although such works significantly extended our knowledge of the motor network implicated in movement planning and execution, we call attention to two main limitations that deserve a critical analysis. 

First, as specified above, most studies accept the idea that connectivity may dramatically change between one task and another. This point is certainly acceptable if one refers to the amount of correlation or mutual information between two signals. Depending on the particular task, in fact, one area can transmit more or less information to another area, conditioned by the level of activities and non-linear phenomena involved. On the other hand, structural causal connectivity (defined as the existence of anatomical connections physically linking brain regions) cannot exhibit such large variations in a task–dependent fashion, and in a brief time scale. In recent studies [15], using a neural mass model to generate reliable signals in an interconnected network, we demonstrated that the estimated functional connectivity can vary dramatically, even in the presence of the same model network, depending on the presence of non–linear phenomena (such as saturation in neural activity). Our idea is that linear models might overestimate the task–dependent component and underestimate the role of structural links, which remain stable across tasks. 

Second, it is well-known that motor execution and motor imagery are based on variations of the neural synchronization in specific brain rhythms, especially in the alpha and beta ranges [16,17]. These rhythms show two main characteristic spatiotemporal patterns during motor processing: a reduction of power in the beta range (event-related desynchronization (ERD)) during motor preparation and performance, which can be considered as a correlate of an activated cortical area [18,19]; an increase in power (event–related synchronization, ERS) during motor suppression, characteristic of a deactivated cortical area or inhibited network [20]. In particular, in patients with stroke, the study of ERD and ERS is important not only to characterize their response but also to define potential motor rehabilitation interventions via brain–computer interface technology [21].

Clearly, the variations in the neural synchronization–which in turn, result in the changes in power in the alpha and beta ranges–are caused by the underlying connectivity between neurons and regions, and determine the consequent motor execution. However, just a few studies focused on the relation between the mechanism underlying this system of rhythms and the motor networks.

The aim of this study is to present a different approach to the problem of model-based connectivity which, although at a preliminary stage (i.e., a proof of concept), can have profound future implications. The distinctive idea here is to ascribe most of the connectivity differences observed between resting state and motor tasks to non–linearity in the neural signal processing, which cannot be grasped by traditional linear models. In other terms, we wish to test the hypothesis that coherence among neuroelectric signals in *different tasks*, as well as ERD and ERS, can be simulated using a *single* connectivity network (i.e., a task-independent network) assuming that the interconnected regions of interest (ROIs) exhibit an oscillatory pattern and that these oscillations are non–linearly transmitted from one region to the other, generating neuronal behaviors supporting different tasks.

Once this network has been estimated, our aim is to summarize brain rhythm power changes during different tasks and their causal relationships into a single theoretical framework, to help understanding the possible underlined neural mechanisms. 

To reach these objectives and provide a proof-of-concept of this approach for motor connectivity, we simulated a network consisting of six ROIs (M1, PMC and SMA in both hemispheres) connected via excitatory and/or inhibitory links. The neuro–electrical activity in each ROI is simulated using a neural mass model (NMM) developed by the authors in past years [22], able to generate multiple rhythms in the alpha, beta and gamma bands. Parameters of the model are assigned to simulate the power spectral densities and coherences among the six ROIs, obtained from electroencephalographic (EEG) data in a patient with unilateral stroke, both in resting condition, and during movement of the affected and unaffected hand. A stroke patient has been chosen to point out whether the method, besides reproducing ERS, ERD and coherence, can also reveal differences in connectivity and in activation in the affected vs. the unaffected hemisphere. 

Finally, we compared our results with the motor networks captured by estimators based on data-driven approaches (temporal correlation, coherence, partial directed coherence and transfer entropy), with the aim to critically discuss the link between task-related functional networks and the network derived from our method based on the NMM, possibly revealing similarities between complementary approaches as well as aspects grasped by the present method not emerging in the functional networks. 

## 2. Material and Methods

### 2.1. Experimental Data: Acquisition, Processing and Connectivity Estimates

#### 2.1.1. Experimental Protocol and EEG Data Measurement

Data subjected to the present analysis were obtained from a previous study [23]. In brief, data were acquired from a stroke patient (male; 62 years; lenticular haemorragia; 1 month since event; left affected hemisphere) enrolled from the rehabilitation hospital ward at Fondazione Santa Lucia, IRCCS, in Rome, Italy. Upon enrolment, the patient was evaluated by means of the European stroke scale (ESS = 75) [24] and the upper limb section of the Fug–Meyer assessment (FMA = 40) [25].

The patient was subjected to a screening session during which electroencephalographic (EEG) signals were recorded during the execution of motor tasks. In particular, the patient was asked to execute a simple movement (sustained grasping movement) with the affected and unaffected hand in separate runs. Each run consists of 30 trials, 15 rest and 15 motor trials in randomized order: the task timing was determined by a movement of the cursor on the computer screen, while during rest trials, the patient was only asked to watch the cursor trajectory. EEG was collected from 61 standard positions (according to the extended 10–20 international system), band pass-filtered between 0.1 and 70 Hz, digitized at 200 Hz, and amplified by a commercial EEG system (BrainAmp, Brainproducts GmbH, Gilching, Germany). 

The experimental protocol described above (see [23] for more details) was approved by the local ethics board of the Fondazione Santa Lucia (Prot.CE/AG4–PROG.244–105) and written informed consent was obtained from the patient.

#### 2.1.2. EEG Preprocessing and Source Reconstruction

In the present study, the preprocessing of EEG signals was performed with Brain–Vision Analyzer (Brainproducts GmbH, Germany). EEG data were downsampled at 100 Hz (with anti–aliasing filter) and bandpass filtered (1–45 Hz). Ocular artifacts were removed by independent component analysis (ICA): the component corresponding to eye blinks was identified and removed by displaying the time course of the ICA components. Lastly, residual artifacts (muscular, environmental, etc.) were removed using a semiautomatic procedure, based on the definition of a voltage threshold (±80 µV).

The preprocessed EEG signals were then segmented, considering the last 4 s of each trial as the period of interest. This procedure was performed for all the runs: rest, motor task performed with the affected hand and motor task performed with the unaffected hand. 

The estimation of the neuroelectrical activity in the brain source space was obtained by solving the linear inverse problem according to the methods described in previous works [26,27,28]. The procedure is based on the use of an average geometry head model (around 8000 equivalent current dipoles, disposed normally to the cortical surface) and allows estimating over time the signed magnitude of the dipolar moment for each cortical dipole. Then, at each time point, the magnitude derived from the centroid of a particular ROI was used as waveform of the cortical activity in that ROI. 

In the present study, we focused on 6 ROIs selected on the basis of their involvement in the tasks under investigation: SMA proper left hemisphere (SMAp L), SMA proper right hemisphere (SMAp R), M1 hand left hemisphere (M1h L), M1 hand right hemisphere (M1h R), dorsal premotor cortex left hemisphere (PMD L), dorsal premotor cortex right hemisphere (PMD R).

#### 2.1.3. Model-Free Network Analysis

In order to gain a preliminary insight into the network structure, we first estimated the connectivity among the 6 ROIs reconstructed in the patient, using four distinct estimators, i.e., the coherence, the temporal correlation, the partial directed coherence and the transfer entropy. It is worth noting that the first two estimators estimate undirected links (i.e., the connectivity from node i to node j is equal to the connectivity from j to i) whereas the last two estimators are directional. Each estimator was applied separately to each experimental condition (rest, movement of the affected hand, movement of the unaffected hand). 

Both *power spectral densities* of individual signals, and the *coherence* between each couple of signals were evaluated using the Welch’s averaged periodogram method, using a Hamming window as long as 0.5 s, 50% overlapping, and zero-padding to have a line spectrum every 0.1 Hz. 

*Temporal correlation* was evaluated by computing the Pearson’s linear correlation coefficient between any couple of signals for each trial and then averaged across trials of the same experimental task. 

The *partial directed coherence* [29] is a linear spectral quantifier which reveals the existence, the direction and the strength of a functional relationship between any given pair of signals in a multivariate data set. In the present work, we used generalized partial directed coherence (gPDC) [30], that modifies PDC to be scale invariant. The optimal order of the multivariate models was estimated by means of Akaike information criterion [31]. 

*Transfer entropy* (TE) is a model free implementation of Wiener’s principle of observation causality [32]. In this work, TE was estimated using Trentool, a software package implemented as a Matlab toolbox under an open-source licence [33]. In particular, we evaluated the so called “high–order” TE, i.e., by considering more than two-time bins for the receiving time series and for the sending time series (the number of past bins used is called embedding dimension and is optimized by the software). 

It is worth noting that all previous methods estimate a pattern of connectivity for each task, providing values that vary depending on the particular task (in this work, basal resting condition, movement of the affected hand and movement of the unaffected hand). Moreover, coherence and gPDC are spectral estimators, hence they provide one value for each frequency; in these two cases, to obtain a single value for each connection, we summarized the values in the beta range. The three task-dependent networks obtained with each method are shown in the Appendix A. However, in order to summarize the main aspects, in section Results only a single network is reported for each method (see Section 3.1), obtained by averaging the three task-dependent networks.

As detailed below (see Section 2.2.2, fitting procedure), the coherence values were exploited in defining initial guesses for model parameters. The other networks mainly serve for comparison with the connectivity network estimated with the proposed method. 

### 2.2. The Neural Mass Model and Model-Based Connectivity Estimation

Our model-based connectivity was evaluated by fitting the outputs of a neural mass model of interconnected ROIs to the experimental data. In particular, we minimized a cost function of the difference between the normalized power spectra of experimental and simulated data, and of the difference between the coherences (see section below). The estimation procedure is designed to provide a unique set of connectivity for the three tasks. 

#### 2.2.1. Qualitative Description of the Neural Mass Model

The model of a single region of interest (ROI) consists of the feedback arrangement among four neural populations: pyramidal neurons, excitatory interneurons, inhibitory interneurons with slow and fast synaptic kinetics. Each population receives an average postsynaptic membrane potential from other neural populations and converts this membrane potential into an average density of spikes fired by the neurons. This conversion is simulated with a static sigmoidal relationship, which reproduces the non–linearity in neuron behavior (the presence of a zone where neurons are silent (below threshold) and an upper saturation, where neurons fire at their maximal activity due to a refractory period).

To model dynamics in a whole ROI, the four populations are connected via excitatory and inhibitory synapses, according to the schema in the upper panel of Figure 1. Each synaptic kinetics is described with a second order system, but with different parameter values. We assumed three types of synapses with different impulse response: glutamatergic *excitatory* assuming that synapses from pyramidal neurons and from excitatory interneurons have similar dynamics; GABAergic inhibitory synapses with *slow* dynamics; GABAergic inhibitory synapses with *faster* dynamics. These synapses are characterized by a gain (*G_e_*, *G_s_*, and *G_f,_* respectively) and a time constant (the reciprocal of these time constants are denoted as *ω_e_*, *ω_s_*, and *ω_f,_*, respectively). The average number of synaptic contacts among neural populations are represented by eight parameters, *C_ij_*, where the first subscript represents the target (post–synaptic) population and the second refers to the source (pre–synaptic) population. 

In order to study connectivity between regions, we assumed that the average spike density of pyramidal neurons of the presynaptic region affects the target region via a weight factor, $W_{j}^{hk}$ (where *j* = *p* or *f*, depending on whether the synapse targets to pyramidal neurons or fast inhibitory interneurons, *h* is the post–synaptic region and *k* the pre-synaptic one) and a time delay, *T*. 

It is worth noting that the synapses $W_{p}^{hk}$ have an excitatory role on the target region *h*, since they directly excite pyramidal neurons (left bottom panel in Figure 1). 

Conversely, synapses $W_{f}^{hk}$, although glutamatergic in type, have an inhibitory role, via a bi-synaptic connection. In particular, both connections ($W_{p}^{hk}$ and $W_{f}^{hk}$) go from the source ROI *k* to the target ROI *h*, but in the inhibitory case the connections are composed of two synapses (from pyramidal neurons in the source ROI *k* to inhibitory interneurons in the target ROI *h* and then from inhibitory interneurons in target ROI *h* to pyramidal neurons still in ROI *h,* see the right bottom panel in Figure 1). Hence, in the following the general terms “excitatory connection” will be used to describe the monosynaptic pyramidal-pyramidal connection, whereas the term “inhibitory bi-synaptic connection” will be used to describe the pyramidal–fast inhibitory–pyramidal connection. 

Finally, each ROI, besides receiving connections from other ROIs, can receives further inputs (represented by Gaussian noise with given mean value and variance) that account for all other external inputs not included in the model. 

In this work, we considered three ROIs in the affected (left L) and unaffected (right R) hemispheres, representing the same ROIs reconstructed from EEG, that is M1h L and M1h R, SMAp L and SMAp R, PMD L and PMD R. We hypothesized that these ROIs can be connected according to the basic scheme shown in Figure 2 (the justification is provided below); the values of these connections were then subjected to the fitting procedure and some of them can go to zero at the end of the fitting. 

The complete set of equations can be found in the Appendix A.

Numerical integration of the differential equations was performed with the Euler integration method, with a simulation step as low as 10^−4^ s. We tested the method’s accuracy by performing some simulations with a much smaller step (hence longer computation times), without observing significant changes in the results. Each simulation lasted 11 s starting from a null initial value of the state variables. The first second of the simulation was then excluded to eliminate the initial transient response. Afterwards, data were passed through a low-pass antialiasing filter, re-sampled at 100 Hz (sampling period 0.01 s) and stored for subsequent processing. In particular, the output signals of the model are the post-synaptic membrane potentials of the pyramidal population in each ROI (i.e., quantity *ν_p_* in Equation (4) of the Appendix A), which is representative of local mean field potential. Finally, computation of power spectrum density and coherence was applied to these model signals. The computation time required to perform one single simulation (including spectra calculation) on a notebook (i7 last generation CPU) was in the range of about 80 s.

#### 2.2.2. Parameter Estimation Method

##### Assumptions on Parameters and Network Topology

We assumed that the following parameters in the model can be assigned on the basis of a fitting procedure between simulated signals and real data:

(1) The 8 coefficients *C_ij_* between the populations (Equations (4), (8), (12) and (18) in Appendix A): These can reflect the number of internal contacts among populations within a ROI. We assumed that they can be different in the left and right hemisphere, even for the same regions, as a consequence of hand lateralization and, above all, of the stroke effect. Hence, we have 8 × 6 = 48 parameters.

(2) The reciprocal of time constants *ω_e_*, *ω_s_*, and *ω_f_* (Equations (2), (6), (10), (14) and (16) in Appendix A). However, to reduce the number of parameters, we assumed that the same regions in the left and right hemisphere have the same time constants. Hence, we have additional 3 × 3 = 9 parameters.

(3) The synapses connecting the different ROIs ($W_{j}^{hk}$ in Equation (19) in Appendix A), providing the model-based connectivity network, and the external inputs ($m_{j}^{k}$ in Equation (19) in Appendix A). In order to identify a possible schema of synaptic connections among the ROIs, we started our analysis looking at data on the motor cortex connectivity in the existing literature [2,5,34]. Then, we assumed that: 

3a—The same regions in the two hemispheres (i.e., M1h L vs. M1h R; SMAp L vs. SMAp R and PMD L vs. PMD R) are connected via inhibitory synapses according to the so-called theory of inhibition [35]. This theory assumes that inhibition occurs between the same function in the two hemispheres to prevent maladaptive cross talk and to allow a given function to become dominant [36,37,38,39].

3b—The connections between regions in the same hemisphere are excitatory. This assumption is strongly supported by previous studies of connectivity in the motor cortex [5,6].

3c—A more difficult problem concerns connections between one area in one hemisphere and a non-homologous area in the other hemisphere. While several studies suggest inhibition [5,6] other suggest that these connections are excitatory, especially when directed toward the performing hand [7,34]. We started from the basic schema depicted in Figure 2, assuming that the connections from one SMAp and the contralateral M1h are inhibitory (this choice reproduces the pattern reported in [5]), whereas the others are excitatory. With just one exception, discussed below, this schema also substantially agrees with the signs obtained from the temporal correlation analysis (see Section 3.1 and Appendix A). In order to test also an alternative hypothesis, the fitting procedure was repeated (alternative fitting procedure), assuming that the synapses from one SMAp and the contralateral M1h are excitatory rather than inhibitory (maintaining unaltered the rest of the schema). Results of the last procedure are reported in Appendix A).

3d—In order to reduce the complexity of the fitting problem, we assumed that M1h L and M1h R receive feedforward synapses from the two SMAps and the two PMDs, but do not send feedback synapses back. This is certainly a strong simplification. As a partial justification, we can observe that the connectivity values estimated with a model-based approach in previous works [5]) exhibit weaker feedback connections from M1 to SMA and from M1 to PMC than the corresponding feedforward connections. However, the fundamental reason to adopt this simplification is to drastically reduce the complexity of the fitting procedure. In fact, thanks to this assumption, we can first assign parameters to the four regions PMD L, PMD R, SMAp L and SMAp R, connected via reciprocal feedback links, and subsequently to assign parameters for the two M1h regions and the feedforward synapses targeting them. In other terms, the problem is split into two sub-problems, resolved in two separated steps of the fitting procedure (Step 1 and Step 2, see below). This simplification can be removed in future works. 

3e—Finally, we assume that the two SMAps and the two PMDs receive an external input (Gaussian noise with a given mean value and assigned variance), impacting on the population of pyramidal neurons. These terms represent all other external sources not included in the model. All input mean values are set to zero in resting conditions, but these values can increase to a positive value during a task execution (left hand or right-hand movement) when the regions are further excited, and may contribute to move the working point of the ROIs along their sigmoidal characteristic outside the central linear region. These 8 values (4 mean values during movement of the affected hand and 4 mean values during movement of the unaffected hand) are additional parameters in the fitting procedure. It is worth noticing that the mean inputs to SMAps and PMDs are the only model parameters that assume different values across the three tasks.

All previous hypotheses will be critically discussed in the last section. 

In conclusion, the total number of estimated parameters for Step 1 and Step 2 of the fitting procedure are:

*Step 1 (PMD L, PMD R, SMAp L, SMAp R):* 32 internal constants *C_ij_*, 6-time constants *ω_ij_*, 4 inhibitory synapses *W_f_*, 8 excitatory synapses *W_p_*, and the 8 input values $m_{p}$. Total: 58 parameters.

*Step 2 (M1h L* and *M1h R):* 16 internal constants *C_ij_*, 3-time constants *ω_ij_*, 4 inhibitory synapses *W_f_*, 6 feedforward excitatory synapses *W_p_*. Total: 29 parameters (in the alternative fitting procedure (see Appendix A), 2 inhibitory synapses *W_f_* and 8 excitatory synapses, *W_p_*, still 29 parameters).

##### Fitting Procedure

In the following we will use the symbols $P_{spe,b}^{ROI}\left(j2\pi f_{k}\right)$, $P_{spe,a}^{ROI}\left(j2\pi f_{k}\right)$, $P_{spe,u}^{ROI}\left(j2\pi f_{k}\right)$ to denote the power spectral densities of the experimental data (subscript *spe*) for a given ROI. The other subscript refers to the basal condition (subscript *b*), movement of the affected hand (subscript *a*) and movement of the unaffected hand (subscript *u*), respectively. *f_k_* is the k–th frequency of the spectrum, computed with the Welch periodogram method. Similarly, we will denote with symbols $P_{\mathrm{mod},b}^{ROI}\left(j2\pi f_{k},\theta \right)$, $P_{\mathrm{mod},a}^{ROI}\left(j2\pi f_{k},\theta \right)$, $P_{\mathrm{mod},u}^{ROI}\left(j2\pi f_{k},\theta \right)$ the spectral densities of the simulated data with reference to the same ROI and at the same frequencies, where *θ* are the model estimated parameters (i.e., these are the power spectral densities of the post-synaptic potential *ν_p_* (Equation (4) in Appendix A) of the given ROI in the model). It is worth noting that, in order to allow a direct comparison between experimental and simulated spectra, and to account for ERD, all spectra have been normalized with respect to the maximum of the spectrum in basal condition in the same ROI. By way of example, by denoting with ${\tilde{P}}_{\mathrm{mod},a}^{ROI}\left(j2\pi f_{k}\right)$ the model spectral density in a ROI during a movement of the affected hand, *without normalization*, we have: (1)$$P_{\mathrm{mod},a}^{ROI}\left(j2\pi f_{k}\right)=\frac{{\tilde{P}}_{\mathrm{mod},a}^{ROI}\left(j2\pi f_{k}\right)}{\max\left\{{\tilde{P}}_{\mathrm{mod},b}^{ROI}\left(j2\pi f_{k}\right)\right\}}$$

A similar normalization has been performed for all power spectral densities, both simulated and experimental. Of course, after these normalizations, all spectra in basal conditions have a maximum as large as 1. ERD is evident during hand movement, both of the affected and unaffected hand, by a normalized spectrum having a maximum smaller than 1. 

As said above, the fitting procedure has been divided in various steps:

(1) Step 0 (preliminary step). First, we estimated a preliminary value to the eight internal parameters *C_ij_* in each ROI, and to the reciprocal time constants, *ω_j_*, in order to simulate the power spectral density in basal resting condition in the range 10–30 Hz. All power densities (experimental and simulated) were previously normalized to their maximum. Fitting was achieved by minimizing the following least square cost function
(2)$$F_{0}(\theta )={\sum_{k}\left(P_{\mathrm{mod},b}^{ROI}\left(j2\pi f_{k},\theta \right)-P_{spe,b}^{ROI}\left(j2\pi f_{k}\right)\right)}^{2}$$
where the sum is extended to all spectral frequencies in the range 10–30 Hz (frequency step Δ*f* = 0.1 Hz), and *θ* is the vector of *internal parameters* in the given ROI. 

In this preliminary step, each ROI is considered separately from the others, i.e., without any connectivity. The only constraint is that we assumed identical *ω_j_* for the homologous ROIs. The estimated parameters are considered as an initial guess for the subsequent steps.

(2) Step 1. Subsequently, we estimated all parameters of the two SMAps and of the two PMDs, including their connectivity weights, according to the diagram in Figure 2. The initial guesses of parameters *C_ij_* in the four ROIs and of the reciprocal time constants, *ω_j_*, were those obtained in Step 0. The initial guesses for the connectivity weights among these four ROIs are described at the end of this section (see below). In this step, we minimized a more complex cost function than in Step 0, to reproduce both the changes in power spectral densities between the three tasks (basal condition, affected hand movement, and unaffected hand movement), and the coherence among the ROIs in basal condition, within the band 10–30 Hz.

By denoting with $C_{spe,b}^{ROI1,ROI2}\left(j2\pi f_{k}\right)$ and $C_{\mathrm{mod},b}^{ROI1,ROI2}\left(j2\pi f_{k},\theta \right)$ the coherences between the signals in ROI1 and ROI2 (with ROI1 ≠ ROI2) computed in basal conditions from the experimental data and from model simulations, respectively, we have:(3)$$\begin{array}{cc} F_{1}(\theta )= & \sum_{tr=b,a,u}\;\sum_{ROI}{\sum_{k}\left(P_{\mathrm{mod},tr}^{ROI}\left(j2\pi f_{k},\theta \right)-P_{spe,tr}^{ROI}\left(j2\pi f_{k}\right)\right)}^{2}+\\ & +\sum_{\begin{array}{l} ROI1ROI2\\ ROI1\neq ROI2 \end{array}}{\sum_{k}\left(C_{\mathrm{mod},b}^{ROI1\;ROI2}\left(j2\pi f_{k},\theta \right)-C_{spe,b}^{ROI1ROI2}\left(j2\pi f_{k}\right)\right)}^{2}\\ & +100\sum_{tr=a,u}\;\sum_{ROI}\left[\max\left\{P_{\mathrm{mod},tr}^{ROI}\left(j2\pi f_{k},\theta \right)\right\}-\max\left\{P_{spe,tr}^{ROI}\left(j2\pi f_{k}\right)\right\}\right] \end{array}$$

The first term in the right hand member represents the square difference of all normalized spectra, computed in the four ROIs and in all trials (basal, affected and unaffected); the second term is the square differences of model vs. experimental coherences, extended to all couples of the four ROIs, computed only in basal conditions; the third term is the differences between the maxima of the spectra, computed in all ROIs, both in the affected and unaffected trials (indeed, since maxima are equal to 1 in basal conditions, this term is zero in the basal case and is not included in the sum). The multiplicative factor, 100, has been introduced so that the last term in Equation (3) has approximately the same weight in the cost function as the first two terms. It is worth noting that the three terms in Equation (3) represent the metrics used to compare model behavior and experimental data (see also the Appendix A). 

(3) Step 2. In this step, we estimated all parameters of the two M1hs, including the feedforward connectivity weights from the SMAps and from the PMDs to the primary motor areas (see Figure 2 again). In this case too, initial guesses of parameters *C_ij_* in these two ROIs and of the reciprocal time constants, *ω_ij_*, were those obtained in Step 0, while the initial guesses for the connectivity weights followed a procedure similar as in Step 1 (see below). The inputs to the feedforward synapses were the spike densities of the previous regions, computed with the optimal parameters estimated in Step 1. The cost function was similar to that used in Step 1 (with the summations on the ROIs extended only to the two M1hs), reflecting both the changes in power spectral densities between the three tasks (basal condition, affected hand movement, and unaffected hand movement), the coherence among the ROIs in basal condition within the band 10–30 Hz, and the maximum of the normalized spectra in the affected and unaffected conditions.

In particular, it is worth noting that in Step 2 the cost function uses only coherence between M1h L and M1h R. The coherences between the M1hs and the SMAps and between the M1hs and the PMDs were never considered in the fitting procedure (either in basal condition or in affected/unaffected hand movement), for the sake of computational simplicity. Hence, all these coherences are posterior predictions. Furthermore, also coherences between all other ROIs in the two motor tasks (with the affected and unaffected hand) are posterior predictions, since only the coherences in basal condition were used in the cost function.

An initial guess for the connection weights among the four ROIs (PMD L and R, SMA L and R) in Step 1 was given on the basis of the coherence values in the beta band. However, since the fitting procedure generally stops at a local minimum, several different initial guesses were produced by adding noise to the parameters (± 50% of its value), so as to obtain several local minima. Then, we chose the best local minimum, i.e., the local minimum which warrants the smallest value of the cost function among those obtained through the previous minimization procedures. 

More particularly, the following iterative algorithm was implemented: 

(1) Starting from the initial set of parameters, established as described above, ten different initial guesses were obtained, by adding noise to the parameter values (uniform distribution in the range ±50% of the parameter values) and the minimization procedure was run for each of these 10 combinations. Each minimization procedure stopped when the cost function did not decrease further, according to a given tolerance. To this end, we set the hyperparameter “Function Tolerance” in the optimization procedure to a value as low as 0.1.

(2) We chose the parameter set that provided the smallest cost function, among those obtained in the previous point. This was named the “present best choice”.

(3) We then added noise to this parameter set (uniform distribution ±50% of the parameter values) thus realizing other 10 combinations of parameters, in the parameter space around the “present best choice”. Minimization was run again for each of these combinations.

(4) The procedure terminated in case the cost function did not decrease at the end of any of the ten minimization runs, compared with that in the present best choice. Hence, the “present best choice” was assumed as the result of the fitting procedure. Otherwise, we updated the new “present best choice” as the one with the smallest cost function and we went back to Point 2. 

A similar procedure was adopted in Step 2, for what concerns the estimation of the weights entering into the M1h L and M1h R. 

All computations were performed using the scientific software environment Matlab (version R2018b Mathworks ^©^, Natick, Ma, USA). Minimization was achieved using a direct pattern search algorithm (Matlab command “pattern search” in the global optimization toolbox), which finds a local solution of the optimization problem without using any information about the gradient of the objective function.

## 3. Results

### 3.1. Model-Free Connectivity Estimation

Figure 3 shows the motor networks obtained with the *coherence* (upper left), the gPDC (upper right), the *TE* (bottom left) and the *correlation coefficient* (bottom right), where the thickness and style of lines reflects the connectivity strength. Specifically, each connection was obtained by averaging the values estimated on the three tasks (the three single networks are displayed in the Appendix A), and we show only connections characterized (on average) by values higher than a given threshold (see legend for more details). Results show how connections *within the hemisphere* are strong between the SMAps and the homolateral PMDs and between the SMAps and the homolateral M1hs (especially in the affected side left), but weak between the PMDs and M1hs. Lateral *interhemispheric connections* are strong between SMAP L and SMAP R, and between PMD L and PMD R, but are negligible between the two M1hs. Finally, we can observe the presence of *interhemispheric connections* between the SMAps and the contralateral M1h with all estimators. Moreover, the gPDC also underlines the presence of connections directed from the SMAps towards the contralateral PMDs (which were not evident with coherence). Compared with the others, TE exhibits a greater number of reentrant connections, which were not evident in the previous estimator: in particular, we can observe connections from the M1h L back to both SMAps. As explained above, we neglected these connections originating from the M1hs when performing the connectivity estimation with the NMM. For what concerns the Pearson correlation coefficient, a fundamental difference compared with other estimators is that its value can be positive or negative, hence, this information can provide a discrimination about the presence of excitatory or inhibitory bi-synaptic connections (see also [15]). A problem, however, is that two signals can have either positive or negative correlation depending on the orientation used during source reconstruction. In other terms, all reconstructed signals have an arbitrary sign. To overcome this problem, we introduced two assumptions in the computation of the correlation coefficients, which allow the choice of the signs, but have no effect on the absolute value: (i) Two homologous areas in different hemispheres inhibit reciprocally, according to the brain inhibition hypothesis (hence, they should have a negative correlation). (ii) Regions in the same hemisphere are connected via excitatory connections (see [5] (hence they should have positive correlation). Starting from these assumptions, and by fixing arbitrarily the sign of one signal (for instance PMD R), we were able to provide a sign for all signals in all ROIs. In particular, we fixed the sign to SMAp R and M1 R to have positive correlation with PMD R, and the sign to all regions in the left hemisphere to have negative correlation with the homologous regions in the other hemisphere. All other correlation coefficients were not fixed a priori, and were obtained a posteriori: this provides a new information on the connection type (excitatory or inhibitory) which could not be obtained using TE, coherence or gPDC estimators (Figure 3 right bottom panel). 

### 3.2. Parameter Estimation with the NMM Model

It is worth noting that the signs of all connections in the right bottom panel of Figure 3 agree with the previous hypotheses (see “assumptions on parameters and network topology” in Section 2.2.2), with the only exception of the connection SMAp L–PMD L which turns out weakly negative. However, we decided to maintain the assumption of positive connection within a hemisphere, which seems more physiological, hence we used an excitatory synapse for this connection during the fitting procedure. Furthermore, the results show that the correlations between each SMAp and the contralateral M1h are negative, suggesting the presence of an inhibitory link (see also Figure 4 in [5]). 

Starting from the general network structure delineated in Figure 2, we estimated all parameters in order to reproduce the normalized spectral densities and coherences in the beta range (see “fitting procedure” in Section 2.2.2). Parameters not subject to fitting can be found in Table 1. These values are the same as in previous works [22,40] and identical for all ROIs. 

Results of Step 1 of the fitting procedure (concerning ROIs SMAp L, SMAp R, PMD L, PMD R) are reported in Table 2 and Table 3, listing the estimated internal parameters of each ROI, and the estimated inputs, respectively, while the estimated connectivity parameters between these ROIs are given in the network diagram of Figure 4, upper panel. It is worth noting that a single parameter set (Table 2 and Figure 4) is used for the three tasks, i.e., the three tasks differ only for the mean input reaching these ROIs (Table 3). Interestingly, the connectivity network resembles the networks estimated with the data-driven methods. 

Furthermore, in order to quantitatively assess the accuracy of our results, we computed the SD of the error, i.e., the SD of the difference between the model results and the real data over the frequency range used for fitting (10–30 Hz). This computation was performed with reference to all the power spectra and coherences during the three tasks. The obtained values are shown in the Appendix A, Appendix A). We consider a good fitting if these SD are below 0.1, and still satisfactory below 0.2. 

Figure 5 shows the normalized power spectral densities in the four ROIs (SMAp L, SMAp R, PMD L, PMD R). The left picture in each panel represents model simulations, while the right picture shows the experimental spectra. First, the model can simulate the ERD observed in each ROI during the two tasks. Second, ERDs are quite different in the two PMDs and in the SMAps. 

A strong desynchronization is evident in both PMDs during the movement of the unaffected hand, especially in the right hemisphere (the one not affected by the stroke). Conversely, desynchronization is less evident during movement of the affected hand, probably as a consequence of stroke. ERD is less evident in the two SMAps, and is just a little stronger during movement of the affected hand. 

In general, the model after fitting is able to reproduce the main aspects of the experimental data in a satisfactory way, as confirmed by the analysis of the SD of errors (Appendix A, Appendix A). We can just observe that the experimental spectrum in the SMAp exhibits a shift to the right during movement of the affected hand (from 20 to about 23–24 Hz) which is much smaller in our model.

Figure 6 shows a comparison between the coherences among the previous four ROIs predicted by the model, and the experimental ones. As it is clear, the model simulates the coherence satisfactorily in the beta range, as confirmed by the analysis of the SD of errors shown in the Appendix A, Appendix A. However, model coherence falls to zero for frequencies above 30 Hz (gamma range) where a strong coherence can still be observed in the experimental data. This difference will be discussed in the last section. 

Table 4 and the lower panel of Figure 4 show the estimated internal parameters of M1h L and M1hR, and the strength of the feedforward connections entering the two ROIs (Step 2 of the fitting procedure), while Figure 7 shows the normalized power spectral densities in the M1h L and M1h R. The accuracy of fitting is good during the movements of the affected and unaffected hands (see the error SDs in Appendix A, Appendix A), but a large difference can be observed for what concerns the basal spectrum in the M1h R (indeed the SD is larger than 0.5). This is essentially due to a shift in the peak of the spectral density at rest in region M1h R: it is at approximately 25 Hz in the model and at about 20 Hz in the experimental data. The peak in the experimental data shifts to about 25 Hz during hand movement, thus explaining the good matching with the model in these conditions. The model simulates the observed ERD. We can observe a significant difference in ERD in the affected hemisphere (M1h L) and in the unaffected one (M1h R). M1h L shows a much stronger desynchronization during movement of the affected hand; conversely, the unaffected region, M1h R, shows a comparable ERD during both movements, although with a moderate prevalence during movement of the unaffected hand. This result (discussed below) may suggest that the unaffected region (M1h R) participates more actively to both movements, compared with the M1h L. F.

Finally, the coherences between the M1h L and the other regions are shown in Figure 8, while the coherences between the M1h R and the other regions in Figure 9. The model can simulate the coherence levels pretty well in the beta range, as demonstrated by the error SDs shown in the Appendix A, Appendix A), despite the fact that the coherences between PMCs and M1hs and between SMAps and M1hs were not used at all in the computation of the cost function in Step 2.

An interesting aspect of the simulation concerns the values of the inputs to the four regions SMAp L, SMAp R, PMD L and PMD R obtained through the fitting procedure (Table 3). These values were set to zero in basal conditions, i.e., all ROIs are working in the central linear region. It is worth noting that, during movement of the affected hand, all regions receive a significant input: this is particularly high to the SMAp R (i.e., in the unaffected side) and in the PMD L (in the affected side). Our interpretation is that both sides participate actively to the task. Conversely, during movement of the unaffected hand, only the regions in the unaffected side (SMAp R and PMD R) receive a strong activation. 

Finally, as described in the Methods section, we repeated the Step 2 of the fitting procedure to test an alternative hypothesis, i.e., assuming that the feedforward connections from each SMAp to the contralateral M1h are excitatory in type (i.e., pyramidal-pyramidal instead of pyramidal–fast inhibitory–pyramidal). The results, reported in the Appendix A, show that the model can simulate the ERD in the M1h regions and coherence rather well also assuming excitatory synapses from the contralateral SMAps. However, fitting is worse than in Figure 7.

## 4. Discussion

In this work, we present an innovative method to build a connectivity model of the brain motor circuits, using oscillatory networks (i.e., neural mass models). The aim was to investigate the problem of rhythms propagation and power spectral density changes (mainly ERD) within the framework of model-based connectivity. The main new aspect of this study compared with former studies is that differences among tasks are not ascribed to context-dependent changes in connectivity, but rather to the effect of non-linearity. This represents a most parsimonious approach to the problem.

The approach is applied to the study of the power spectral densities and coherence in the beta band in a single subject after stroke, both in resting conditions and during movement of the affected and unaffected hands. Results show that a single set of parameters can mimic all these conditions quite well, and that the values of connectivity obtained are in qualitative agreement with those obtained in former studies. The present study is a proof of concept, applied to a single patient. Indeed, since the connectivity network after stroke may differ significantly among individuals, as a consequence of the locus of the lesion and of time after stroke, each method for connectivity estimation must be applied to single cases. We do not aspire to present a statistic among several subjects here (i.e., a group analysis), but to show how a single fitting procedure actually works.

Results of our study agree quite well with some results appeared in the literature, in which the motor network was assessed in relation to neuroimaging data. First, our main network structure (Figure 4) shares its basic aspects with that obtained by [5] on normal individuals. These authors suggest that the most prominent positive influence on intrinsic M1 activity is exerted by the ipsilateral SMA, whereas the intrinsic coupling between PMC and ipsilateral M1 is less pronounced. Moreover, the majority of transcallosal pathways exerts a negative influence on the activity of motor areas in the contralateral hemisphere. Additionally, the interhemispheric interaction between the M1 areas has been studied in humans by means of TMS too [41]. These experiments suggest that both M1 exhibit a mutual inhibitory influence on each other [42,43]. The presence of transcallosal inhibition agrees with the theory of interhemispheric inhibition [44]. In this theory, the capacity of one ROI in a hemisphere to accomplish a specialized task results from effective suppression of the congruent activity in the other hemisphere. This kind of interhemispheric interaction has been observed not only in motor tasks (as in the present study) but also in language and non-spatial visual processing tasks [45].

While the motor network in healthy individuals appears quite symmetrical, a significant asymmetry can be observed in patients after stroke (both in the acute and chronic stages), since neural coupling among areas can be dramatically altered. In particular, a typical finding in stroke connectivity studies [1,34,46,47] is a decreased connectivity in the perilesional area, i.e., a reduction of positive influences from ipsilesional SMA and PMC onto ipsilesional M1, which is observed shortly after the insult and slowly resolves with time. Previous studies showed a correlation between inter-hemispheric coupling in the beta and gamma bands and corticospinal tract integrity [48] as well as between ipsilesional connectivity after a BCI–assisted intervention and the consequent functional recovery in the same bands [23]. This finding agrees with the network connectivity shown in Figure 4. Furthermore, in several cases the intrinsic connections *C_ij_* are higher in the R hemisphere compared with the L one (Table 2 and Table 4). 

Moreover, a number of functional neuroimaging studies have shown that, in stroke patients, movements of the affected hand evoke higher and more extended neural activity in cortical brain regions [2,8,49,50,51,52]. In particular, unilateral movements of the affected limb are associated with a more bilateral activation pattern in primary motor and premotor areas as compared to neural activity assessed during unilateral hand movements in healthy subjects [46,49,51,52]. The latter results too agree with our model predictions. In fact, our model ascribes the observed differences between the affected and unaffected hemispheres to the following main mechanisms: (i) a reduction in the feedforward connections from the premotor areas (both SMA and PSD) to the M1 in the affected side (see Figure 4), in agreement with previous data; (ii) a reduction of cross-lateral inhibition from the affected to the unaffected side (see Figure 4), especially evident in the cross-talk between the two SMAs and the two PMDs. The latter mechanism implies a disinhibition of the unaffected side during movement of the paretic limb; (iii) a significant asymmetry in the inputs reaching the PMDs and the SMAs during the two hand movements. In particular, just the unaffected hemisphere is strongly stimulated by external excitatory inputs during movement of the unaffected hand (see the third column of Table 3), resulting in greater activation in that hemisphere only. Conversely, both sides receive significant stimulation during movement of the affected hand (Table 3 second column). This wider input excitation during movement of the affected hand, together with less inhibition from the affected to the unaffected side, results in a significant activation in both hemispheres and in a more bilateral excitation pattern, as suggested by the previous literature. Hence, the areas in the contralesional hemisphere seem to be behaviourally important in the reorganized motor network, to facilitate movements of the affected hand. 

The presence of external inputs only to the SMAs and PMCs also agrees with previous studies. Indeed, these areas can receive connections from the visual areas as well as from the parietofrontal system [2]. Finally, the strong input predicted in our study to the PMD R (i.e., in the unaffected side) during movement of the affected hand also agrees with some TMS studies [53,54] showing that interfering with activity of contralesional dorsolateral premotor cortex is associated with a decline in motor performance in stroke patients, but not in controls.

Furthermore, some considerations emerge if we compare the connectivity network obtained with the present model, with the networks obtained using model-free estimation techniques (Figure 3). We can observe that the method we here propose can grasp the main changes reported in the literature between the affected and unaffected hemisphere, both for what concerns a reduction in the connectivity in the affected side, and the reduced transcallosal inhibition from the affected to the unaffected side; these changes clearly emerged in the proposed method but are not equally captured by the other estimators. 

There is only an important aspect in Figure 4 which seems at odd with present knowledge and deserves a discussion, i.e., the strong *inhibitory* connection coming from the affected SMA to the unaffected M1. In particular, [34] observed that the negative coupling from ipsilesional SMA and ipsilesional PMC on contralesional M1 is significantly reduced in stroke patients. These differences in connectivity may be the consequence of some limitations in the model or in the fitting procedure (see below). However, we think there are more probable other reasons. A fundamental cause of differences between our model and previous studies is that our model provides a single set of connectivity values, which simulate all tasks together, whereas previous methods based on neuroimaging data make use of a variable connectivity matrix. Finally, discrepancies can also depend on differences in behavior among a rhythmic model, which simulates oscillatory patterns (particularly in the beta band) and models fitted on static neuroimaging data.

The latter consideration moves our analysis to the comparison with the results of more modern techniques, which make use of causal models to simulate EEG/MEG data, with emphasis on power spectral density, brain oscillations and non-linear coupling. 

Two main approaches can be found. In the first [55] the model describes phenomenologically the evolution of spectral densities in multivariate time-series including coupling parameters both within and between frequencies, thus taking also non-linear coupling effects into account. This class of models differs significantly from the present since spectra are not simulated with a biologically inspired model and a matrix of parameters is introduced to encode the task dependent influence [56]. Using this approach, Chen et al [57]. studied the human motor system during hand grip, and reached the conclusion that the task-dependent motor network is asymmetric during right hand movements and exhibits strong evidence for nonlinear coupling.

More similar to the present approach, is the recent proposal to use biologically inspired neural mass models within the framework of effective connectivity estimation (Friston et al., 2019), in order to simulate EEG/MEG data [58,59,60,61]. A fundamental difference is that in these models the intrinsic and extrinsic connectivity parameters are affected by the inputs (i.e., are context dependent, in particular see [58]) while the approach we propose here is based on context-independent connectivity and on the effect of non-linearity. Moreover, it is worth noting that the previous models were not directly applied to the motor network and hand movements, hence we cannot compare their results with ours. 

Our study, besides reproducing ERD quite well during different tasks and in different ROIs, provides also some indications on the possible underlying neurobiological mechanisms. Although ERD is a well-known phenomenon, whose first description can be dated back to the mid-seventies [62], its modeling interpretation is still controversial. It is generally thought that event-related desynchronization in the beta range represents increased sensorimotor cortex excitability [63,64]. An important aspect to be stressed is that the possibility in our model to simulate different tasks and ERD with a single connectivity pattern depends on the presence of non-linear terms, and specifically on the presence of sigmoidal relationships. Non-linearity makes the spectra and also the amount of information transmitted from one region to another different among trials. This occurs as a consequence of a shift of the working point in the sigmoidal relationship, which computes the population firing rate in terms of the excitatory potential. In particular, if the sigmoidal relationship was replaced with a linear one: (i) the model would become linear, hence it could not simulate self-sustained rhythms in the form of limit cycles; (ii) the model would just amplify or attenuate the white noise at different frequencies, as in classic autoregressive linear models (i.e., linear filters), and the spectra would not change significantly with the task (provided that power density of input noise does not change and that connectivities are kept constant).

Looking at our simulations, we can provide the following neurobiological hypothesis for ERD. When populations work in the central regions of the sigmoidal relationship, they exhibit a good capacity to modify the spiking frequency of individual neurons in response to small changes in the input potentials. This corresponds to a high ability of neurons to synchronize their relative activity, resulting in large collective oscillations. Conversely, when populations work in the upper saturation region (which, as to real spiking neurons, corresponds to the presence of a refractory period) the spiking frequency of individual neurons can only be moderately affected by changes in the input potential. We claim that, in this situation, neurons lose the capacity to synchronize themselves (since synchronization requires the possibility to adapt their reciprocal phase hence to modify their spiking period) thus resulting in a high average activity but with minor oscillatory waves. It is worth noticing, indeed, that in our simulations, beta band ERD in ROIs is in general associated to a decrease in beta-band coherence (see Figure 5, Figure 6, Figure 7, Figure 8 and Figure 9) compared to resting state. 

However, as underlined by [65] we are aware that in this class of NMM models the sigmoidal formulation does not derive from a comparison with a microscopic neuron dynamic, i.e., with a biophysical description of spiking neurons, hence our interpretation is just speculative and requires further study.

Recently, Byrne et al. [65,66] proposed the use of a different kind of neural mass models, which directly incorporates a description of the population synchrony. With this model, the authors simulated the changes on power spectral density of the motor cortex during movement. In particular, they observed that an increase in the excitatory drive causes a decrease in the oscillatory amplitude, i.e., a desynchronization. This result was supported with data obtained with a high dimensional spiking network. 

It is worth noting that, although our model and that proposed by Byrne et al [66] are quite different, they both predict a decrease in oscillation amplitude in response to large drive. Further studies are necessary to compare these two classes of models, and with the behavior of spiking network models.

A further interpretation of ERD, which resembles the present one, was proposed by [67] using NMM consisting of two excitatory-inhibitory populations in feedback. ERD was mimicked through variations of the external excitation, together with a change in the connection strength between excitatory and inhibitory populations attributed to short-time plasticity. Again, it is worth noting that we do not consider the latter context-dependentmechanism. More similar to our approach, [68] simulated ERD with a model consisting of two neural mass models connected together, to reproduce the transmission of information from one cortex to the other. As in our study, cortical activation or deactivation can move the working point in the upper saturation region, causing ERD, or in the linear region, causing ERS. 

Finally, we wish to emphasize some limitations of the present study and point out lines for future research. 

A first important limitation consists in the difficulty to find a minimum of the cost function, and so in the computational complexity of the fitting procedure. As it is well known, results of a non-linear cost function minimization significantly depend on the initial guess (i.e., on the initial value assigned to the parameters). Unfortunately, the use of global estimation techniques (to reach an absolute minimum instead of a local minimum) would be computationally too cumbersome. 

Second, the present model can simulate the power spectral densities and coherences quite well in the beta range, but does not simulate coherences in the gamma band (above 30 Hz). There are two possible explanations for this aspect. One possibility is that a significant gamma rhythm is received from other areas not included in the model [40,69]. In alternative, it is possible that a more complex neural mass model, or a different combination of parameters, would allow a better simulation of both gamma and beta coherences together. However, the model used in this paper was built to simulate both beta and gamma rhythms together (see [22]), and makes use of fast inhibitory interneurons often neglected in other NMMs studies. Hence, we do not think that gamma band limitation can be ascribed to limitation of the NMM used. 

Here, we focused on a single patient to show a proof–of–principle of the method we propose. The analysis of many cases, or the longitudinal study of the same subject vs. time, could exploit this new approach to address specific neuroscientific questions, and may be performed in further study, using a better automatization of the fitting procedure. More specifically, although neural mass models have been largely corroborated in previous studies ([22,40,70,71]) we are aware that the proposed method for connectivity assessment still deserves a thorough validation. Indeed, an accurate validation in future work will require an analysis on both healthy subjects and patients with stroke lesions. Moreover, longitudinal changes of connectivity network in patients should be assessed at different time periods after stroke. This analysis will be essential to validate the present “proof of concept” thus supporting: (i) the possibility to explain different tasks with a single network in several different cases; (ii) the possibility to grasp differences between connectivity networks in healthy subjects and in patients after stroke; (iii) the possibility to understand the reorganization of a network with time evolution after stroke. All these validation points deserve future application of the present algorithm on a larger data set.

## 5. Conclusions

In conclusion, the present study shows that many different aspects of brain rhythms in the motor network (including power spectra changes, ERD, coherences in the beta range) can be simulated quite well in different tasks, using a single set of parameters for inter–region and intra-region connectivity, without the need to assume a task-dependent change in connectivity weights. Moreover, our study provides a neurobiological interpretation of ERD and ERS, in term of the working point on the sigmoidal relationship. 

Our results suggest that, in the patient here analysed, the contralesional PMD and SMA (PMD R and SMA R) are important in motor reorganization during movement of the affected hand; they receive strong external input and a smaller inhibition from the affected side, and send stronger excitation to the other hemisphere. During movement of the affected hand, ERD occurs both in M1hL and (although to a less extent) in M1hR. This may suggest that M1hR contributes to perform the movement with the affected hand or that the affected hemisphere exerts a reduced inhibition on the healthy one. The observed changes between tasks are due to differences in the external inputs and to non-linear phenomena, but not to task-dependent changes in connectivity. This may represent an important novel aspect to be considered in future studies of brain connectivity.

## Figures and Tables

**Figure 1 brainsci-11-01479-f001:**
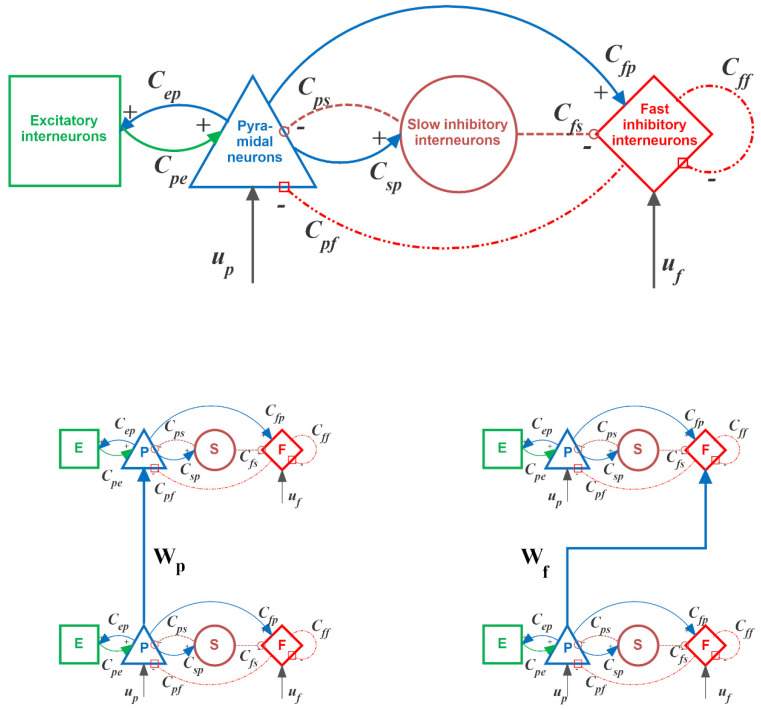
Block diagram of the neural mass model (upper panel) used to simulate activity in a single region of interest (ROI). Continuous lines denote excitatory synapses (from pyramidal neurons, blue lines, or from excitatory interneurons, green lines), magenta dotted lines denote slow inhibitory synapses, and red dash-dotted lines denote fast inhibitory synapses. The bottom panels show two exempla of connections among ROIs: *excitatory* (pyramidal-pyramidal) in the left and bi-synaptic *inhibitory* (pyramidal–fast inhibitory–pyramidal) in the right.

**Figure 2 brainsci-11-01479-f002:**
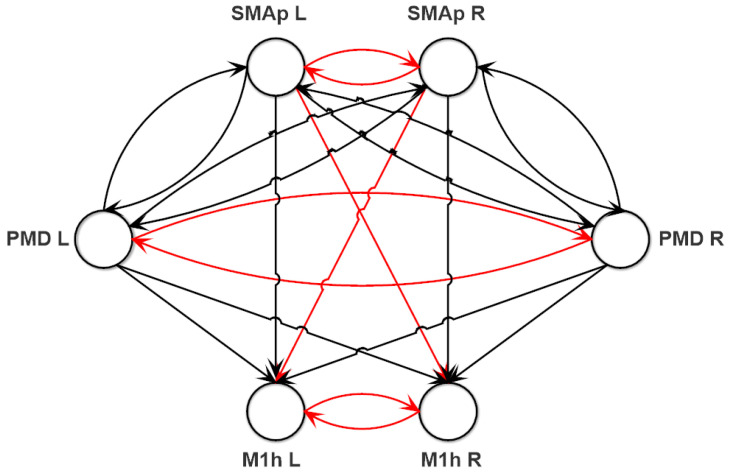
General structure of the connections used to fit the neural mass model to the experimental data. It is evident the presence of feedback connections between the SMAps and the PMDs, and the presence of feedforward connections towards the M1hs. Black lines denote excitatory pyramidal-pyramidal connections, whereas red lines denote inhibitory bi-synaptic connections (pyramidal–fast inhibitory–pyramidal).

**Figure 3 brainsci-11-01479-f003:**
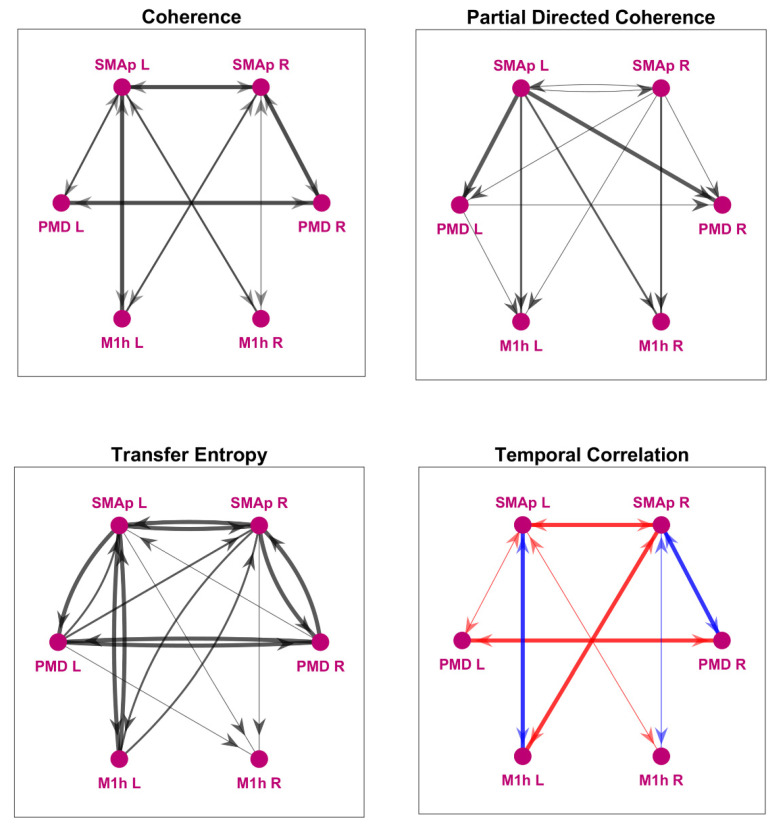
Functional connectivity networks obtained using model-free connectivity estimation methods. All the displayed networks are obtained by averaging the estimates over the three tasks. *Coherence* (upper left panel) is a symmetrical quantity (i.e., it is the same in both directions), therefore, all connections are bidirectional: only connections with maximum coherence value (averaged over the tasks) above 0.3 within the beta band are shown. Thick lines denote a maximum coherence greater than 0.5, medium lines a maximum coherence in the range 0.4–0.5, and thin lines a maximum coherence in the range 0.3–0.4.; gPDC (upper right panel) is a directional quantity: only connections with a gPDC value higher than 0.01 (averaged over the three trials) within the beta band are shown. Line thickness is proportional to the mean gPDC level (thin lines: gPDC < 0. 1; medium lines: 0.1 < gPDC < 0.3; thick lines: gPDC > 0.3). *TE* (bottom left panel) is also a directional quantity: only connections with TE value (on average) higher than 20% of the maximum TE (maxTE = 0.023) are shown. Line thickness is proportional to the mean TE level (thin lines: TE between 20% and 30% of maxTE; medium lines: TE between 30% and 50% of maxTE; thick lines: TE greater than 50% of maxTE). *Pearson’s linear correlation* (bottom right panel) is a symmetrical quantity; therefore, all connections are bidirectional: only connections with correlation coefficient absolute value above 0.3 are shown. Here, colors specify whether the bidirectional connection is *excitatory* (blue, positive correlation coefficient) or *inhibitory* (red, negative correlation coefficient). It is worth noting that the sign of the correlation never changed from one task to another. Thick lines denote a Pearson correlation coefficient greater than 0.5, medium lines a correlation coefficient in the range 0.4–0.5, and thin lines a correlation coefficient in the range 0.3–0.4.

**Figure 4 brainsci-11-01479-f004:**
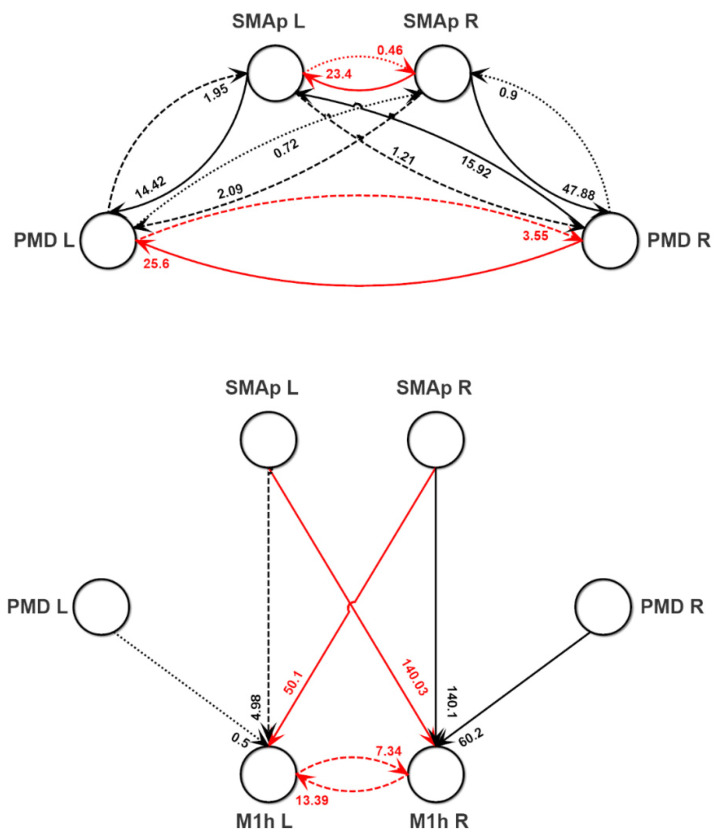
Connectivity strengths obtained by fitting the neural mass model to the normalized power spectra and coherence of the experimental data (see Section 2.2.2)). Since the fitting procedure has been divided in two main steps (Step 1 and Step 2), results of Step 1 (concerning the SMAp L, SMAp R, PMD L, PMD R) are reported in the upper panel, while results of Step 2 (concerning the connection strengths entering into the M1h L and M1h R) are reported in the second panel, although a single network should be considered in the reality. Black lines denote excitatory pyramidal–pyramidal connections, whereas red lines denote inhibitory bi-synaptic connections (pyramidal–fast inhibitory–pyramidal). Continuous lines are used to denote the higher synapses, dashed lines intermediate synapses, and dotted lines the smaller synapses. All remaining synapses are set at zero. The other parameters of the fitting procedure (internal constants within each ROI and inputs to the SMAps and PMDs) can be found in Table 2 and Table 3.

**Figure 5 brainsci-11-01479-f005:**
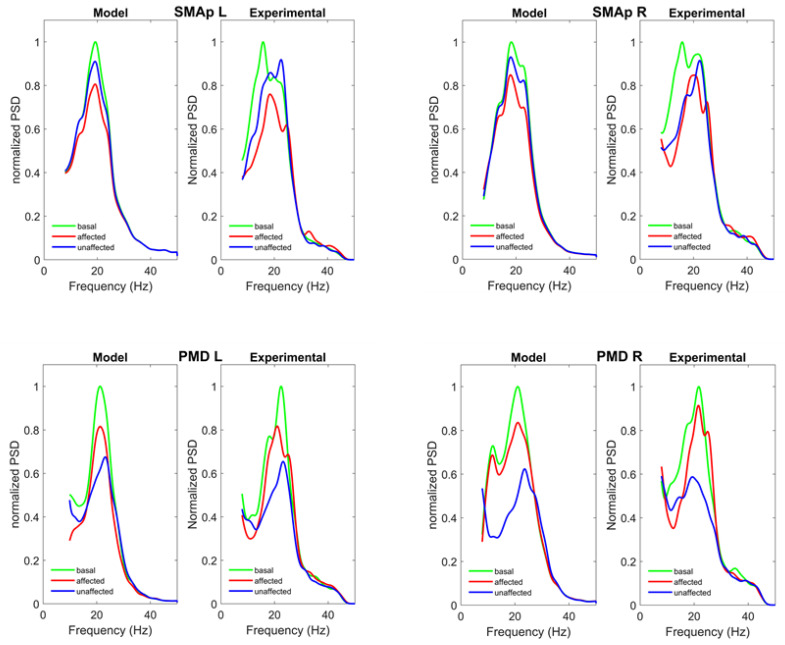
Normalized power spectral densities in the SMAp L (upper left panel), SMAp R (upper right panel), PMD L (bottom left panel) and PMD R (bottom right panel) obtained in basal condition (green lines) and during movement of the affected (red line) and unaffected (blue lines) hands. In each panel, the left part represents model simulation results with optimal parameter values, and the right part the spectra computed from the experimental data. The model after fitting is able to reproduce the main aspects of the experimental data in a satisfactory way, as confirmed by the analysis of the SD of errors (Appendix A, Appendix A).

**Figure 6 brainsci-11-01479-f006:**
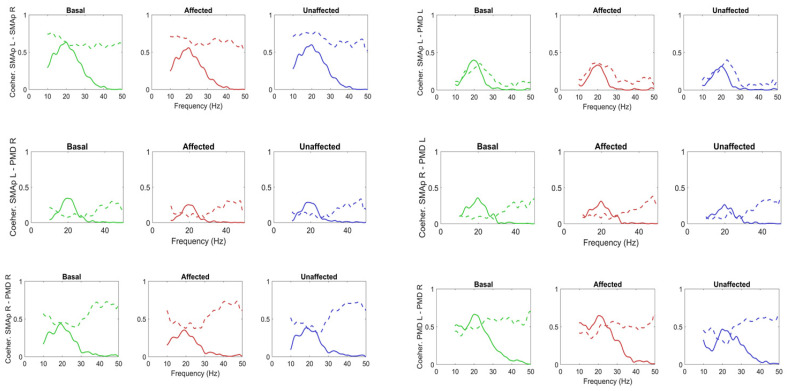
Coherences among the SMAp L, SMAp R, PMD L and PMD R obtained in basal conditions (green lines) and during movement of the affected (red lines) and unaffected (blue lines) hands. The continuous lines represent model simulation results with optimal parameter values, and the dashed lines the values computed from the experimental data. The model can simulate coherences in the range 10–30 Hz quite satisfactorily in all conditions (as shown by SDs of errors in Appendix A, Appendix A), but does not incorporate coherence in the gamma range (>30 Hz). It is worth noting that only the coherences in basal conditions were used in the cost function of the fitting procedure. The others are posterior predictions.

**Figure 7 brainsci-11-01479-f007:**
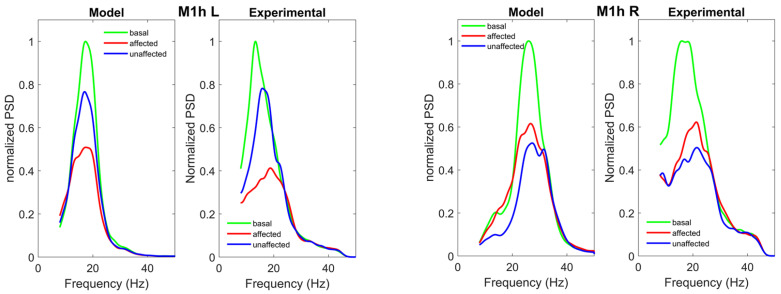
Normalized power spectral densities in the M1h L (left panel) and in the M1h R (right panel), obtained in basal condition (green lines) and during movement of the affected (red lines) and unaffected (blue lines) hands. In each panel, the left figure represents model simulation results with optimal parameter values, and the right figure the spectra computed from the experimental data. The model can simulate the power spectral density, and ERD in a satisfactory way in all conditions, as shown by SDs of errors in Appendix A, Appendix A), with the exception of the basal M1h R.

**Figure 8 brainsci-11-01479-f008:**
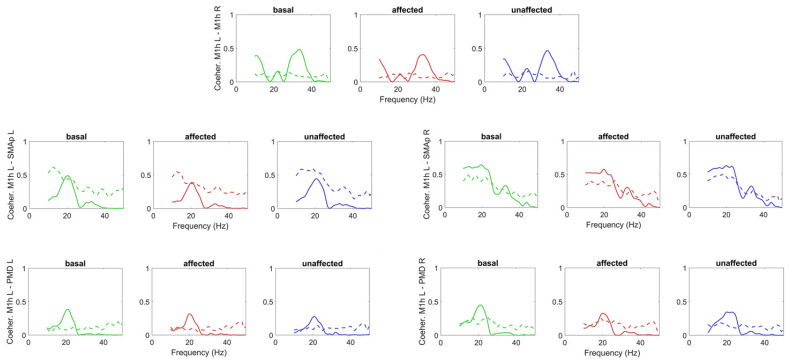
Coherences among the M1h L and all other ROIs obtained in basal conditions (green lines) and during movement of the affected (red lines) and unaffected (blue lines) hands. The continuous lines represent model simulation results with optimal parameter values, and the dashed lines the values computed from the experimental data. It is worth noting that only the coherence M1h L–M1hR in basal conditions was used in the cost function of the fitting procedure. All the others are posterior predictions.

**Figure 9 brainsci-11-01479-f009:**
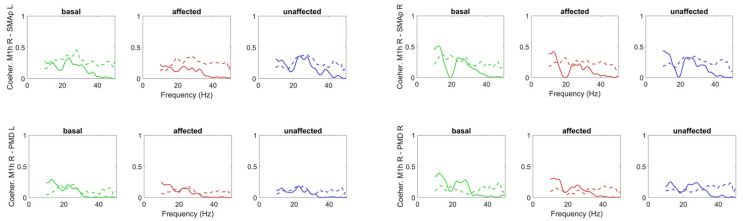
Coherences among the M1h R and all other ROIs obtained in basal conditions (green lines) and during movement of the affected (red lines) and unaffected (blue lines) hands. The continuous lines represent model simulation results with optimal parameter values, and the dashed lines the values computed from the experimental data. It is worth noting that these coherences were not used in the cost function of the fitting procedure, hence are posterior predictions.

**Table 1 brainsci-11-01479-t001:** Parameters assumed fixed for all tasks.

*Parameter*	Value	Meaning
*e* _0_	2.5 Hz	Saturation of the sigmoid
*r*	0.56 mV^−1^	Parameter related with the central slope of the sigmoid
*T*	16.6 ms	delay
*G_e_*	5.17 mV	Synaptic gain excitatory
*G_s_*	4.45 mV	Synaptic gain inhibitory slow
*G_f_*	57.1 mV	Synaptic gain inhibitory fast

**Table 2 brainsci-11-01479-t002:** Internal parameters estimated on ROIs: SMAp L, SMAp R, PMD L and PMD R. It is worth noting that ω are the same for the two SMAps and for the two PMDs.

Parameter	SMAp L	SMAp R	PMD L	PMD R	Meaning
ω_e_	76.14 s^−1^	76.14 s^−1^	62.97 s^−1^	62.97 s^−1^	Reciprocal of a time constant
ω_s_	33.95 s^−1^	33.95 s^−1^	24.07 s^−1^	24.07 s^−1^	
ω_f_	336.8 s^−1^	336.8 s^−1^	734.9 s^−1^	734.9 s^−1^	
*C_ep_*	34.90	5.55	47.41	26.26	Internal connectivity constant
*C_pe_*	12.02	5.46	29.04	50.73	
*C_sp_*	13.94	53.58	78.70	227.61	
*C_ps_*	6.92	53.98	68.80	123.99	
*C_fs_*	10.38	5.25	18.52	4.62	
*C_fp_*	45.02	40.91	80.80	55.06	
*C_pf_*	39.06	28.36	34.24	72.65	
*C_ff_*	22.83	5.67	5.44	4.74	

**Table 3 brainsci-11-01479-t003:** Mean value of the external excitatory inputs estimated on ROIs: SMAp L, SMAp R, PMD L and PMD R during the three tasks (values at rest were set to zero).

Parameter $m_{j}^{h}$	Rest	Movement Affected Hand	Movement Unaffected Hand	Meaning
$m_{p}^{SMAP\;L}$	0	24.66	0	Input mean value to a ROI
$m_{p}^{SMAP\;R}$	0	190.29	111.87	
$m_{p}^{PMD\;L}$	0	277.28	0	
$m_{p}^{PMD\;R}$	0	21.09	482.99	

**Table 4 brainsci-11-01479-t004:** Internal parameters estimated on ROIs: M1h L, M1h R, during the first fitting procedure, i.e., assuming that the connections between each SMAp and the contralateral M1h are inhibitory in type. It is worth noting that ω are the same for the two ROIs.

Parameter	M1h L	M1h R	Meaning
ω_e_	60.78 s^−1^	60.78 s^−1^	Reciprocal of a time constant
ω_s_	68.24 s^−1^	68.24 s^−1^	
ω_f_	689.50 s^−1^	689.50 s^−1^	
*C_ep_*	176	64	Internal connectivity constant
*C_pe_*	63	56	
*C_sp_*	172	329	
*C_ps_*	114	116	
*C_fs_*	20	20	
*C_fp_*	44	204	
*C_pf_*	68	60	
*C_ff_*	36	20	

## Data Availability

The neural mass model and the core of the minimization procedures have been implemented in Matlab. Codes are available in the repository ModelDB at http://modeldb.yale.edu/267174, accessed on 1 November 2021.

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
