# Peer review of "A Novel Method to Assess Motor Cortex Connectivity and Event Related Desynchronization Based on Mass Models"

_brainsci, 2021, doi:10.3390/brainsci11111479_

Round 1
Reviewer 1 Report
The authors present a model-based method to estimate task independent motor connectivity. They use six cortical regions of interest ROIs involved in hand movement and a neural mass model to reproduce the oscillatory activity dynamics of each region. As an input, the parameters of the model have been assigned both power spectral densities and coherences of a patient with left-hemisphere stroke during resting conditions, movement of the affected and movement of the unaffected hand. The presented model can simulate the three conditions using a single set of connectivity parameters, assuming that only inputs to the ROIs change from one condition to the other and without relying on a task-dependent connectivity.
The idea of a task-free connectivity model is very interesting. The data, hypothesis and methods implementations are appropriate. The paper is very well written and very clear. I have some minor remarks and some questions in the following.
-General question: I have a major concern regarding the validation of the model. The sentence in the discussion paragraph “Results of our study agree quite well with some results appeared in the literature” is not sufficient to validate the model.
-In Material and method, the authors did not mention what tools were used for EEG signals processing. They cite ICA but we don’t know how ICA was used. Did they record EOG and ECG simultaneously and correlate the signals with ICA components or reject ICA component based on a threshold value?
-Line 203 punctuation, split the sentence after Materials II to make it more comprehensible
-Line 440. What do you mean by the best local extremum? How ‘best ‘is quantified?
I have concerns regarding the minimization criteria of the cost function of the fitting procedure. The authors state that it is based on a visual inspection. I suppose that the procedure is iterative and then it would be more appropriate to set a convergence criterion rather than deciding about the convergence with visual inspection. This part needs to be described more clearly
Author Response
See the enclosed file

Reviewer 2 Report
This paper presented a neural mass model for the brain motor circuits. The model was novel that it didn't assume context-dependent changes in connectivity. It was applied to a single stroke patient's data during movement of the affected and unaffected hands. From the estimated connectivity, the authors were able to find evidences that corresponds to the previous research, which were not present in the analyses from estimating the model-free connectivity. The paper is well-presented with a lot of details. Although it is only a single subject study, it is still pretty interesting considering it is a proof-of-concept. I only have some minor comments below,
- In Figure 5 where the authors mentioned that the model is able to fit the experimental data "very well", how is it determined to be a good fit? Is there any error metric that can be computed? It also seems that there is some spectrum shift in SMAp between the model estimation and the experimental data.
- Similarly for Figure 6 and other similar figures like 8 and 9, how did the authors determine that the fit in the beta range was "quite well"?
- It seems that the innovation in this paper is the usage of the sigmoidal function. I am wondering if the author could provide some insight into the alternative results that will be obtained if we switch to the linear function. How worse could it be?
Author Response
See enclosed file
